# Concise Review: Bioengineering of Limbal Stem Cell Niche

**DOI:** 10.3390/bioengineering10010111

**Published:** 2023-01-12

**Authors:** Mohammad Soleimani, Kasra Cheraqpour, Raghuram Koganti, Seyed Mahbod Baharnoori, Ali R. Djalilian

**Affiliations:** 1Eye Research Center, Farabi Eye Hospital, Tehran University of Medical Sciences, Tehran 1336616351, Iran; 2Department of Ophthalmology and Visual Sciences, University of Illinois at Chicago, Chicago, IL 60612, USA

**Keywords:** limbal stem cells, limbal stem cell deficiency, LSCD, limbal stem cell niche, limbal niche, bioengineering, niche restoration

## Abstract

The corneal epithelium is composed of nonkeratinized stratified squamous cells and has a significant turnover rate. Limbal integrity is vital to maintain the clarity and avascularity of the cornea as well as regeneration of the corneal epithelium. Limbal epithelial stem cells (LESCs) are located in the basal epithelial layer of the limbus and preserve this homeostasis. Proper functioning of LESCs is dependent on a specific microenvironment, known as the limbal stem cell niche (LSCN). This structure is made up of various cells, an extracellular matrix (ECM), and signaling molecules. Different etiologies may damage the LSCN, leading to limbal stem cell deficiency (LSCD), which is characterized by conjunctivalization of the cornea. In this review, we first summarize the basics of the LSCN and then focus on current and emerging bioengineering strategies for LSCN restoration to combat LSCD.

## 1. Introduction

The cornea is the transparent structure of the anterior eye and has several critical roles, including separating the inner parts of the eye from the outer environment and properly transmitting light to be focused on the retina. The most superficial layer of the cornea is the epithelium, which is composed of nonkeratinized stratified squamous cells and has a significant turnover rate. The junction between the cornea and the adjacent conjunctiva is an annular transition zone referred to as the limbus [1]. Limbal integrity is vital to maintain the clarity and avascularity of the cornea as well as regeneration of the corneal epithelium. Limbal epithelial stem cells (LESCs) are located in the basal epithelial layer of the limbus and preserve this homeostasis. LESCs show multiple markers, such as K5, K14, K15, Vimentin, Notch-1, TXNIP, ABCB5, and ABCG2, which can help to isolate and identify them [2]. Proper functioning of LESCs is dependent on a specific microenvironment, known as the limbal stem cell niche (LSCN), which demonstrates specific physical, autocrine, and paracrine functions. This structure is made up of various cells, an extracellular matrix (ECM), and signaling molecules. Different etiologies may damage the LSCN, leading to limbal stem cell deficiency (LSCD), which is characterized by conjunctivalization of the cornea [3,4]. A proper understanding of limbal ultrastructure, the limbal microenvironment, and functions of LESCs is fundamental to generating LSCN restoration strategies [5]. In this review, we first summarize the basics of the LSCN and then focus on current and emerging bioengineering strategies for LSCN restoration to combat LSCD.

## 2. Limbal Niche (LN)

### 2.1. Stem Cell Niche

Generally, stem cells require particular anatomical sites for preservation and proper functioning [1]. These microenvironments are termed the stem cell niche and contain several components in addition to stem cells, such as supportive cells, several signaling factors, neurovascular inputs, and an ECM. This niche plays a critical role in the terminal differentiation of stem cells into intended tissue cells [2]. While a significant number of cells have the potential to act as stem cells, only a small fraction of them accomplish this task [3].

The niche is critical to limbal stem cell functioning. In one study, total removal of the limbal epithelium with a spared niche was compared to simultaneous injury of the limbal epithelium and niche. In the first group, the epithelium recovered, while the latter group demonstrated corneal neovascularization without healing [4]. Pure injury to the niche without involvement of LESCs may arrest wound healing upon subsequent injury to the limbus [5].

### 2.2. LN Microstructure and Components

The LSCN is located in the limbal crypts formed from fibrovascular ridges, called the palisades of Vogt [6] (Figure 1). These structures have a length of 0.31 mm and a width of 0.04 mm and are typically more detectable on the superior and inferior sections of the cornea compared to the nasal and temporal regions [7]. Limbal epithelial crypts and focal stromal projections are the other compartments of this area, which promote signal integration from different factors of the niche [6,8]. Limbal epithelial crypts are projections from the undersurface of the limbal epithelium into the stroma. These structures could be parallel or perpendicular to the palisades of Vogt. Focal stromal projections are finger-shaped projections of the stroma containing a central blood vessel, which extend upward into the limbal epithelium [6,8]. Notably, these structures are specific to pigs and humans but no other mammals [9,10]. Multiple cell types, such as nerve cells, vascular cells, immune cells, mesenchymal cells, and melanocytes, are detected in the stroma of the limbus [11]. Melanocytes produce melanin to protect LESCs against UV radiation and scavenge reactive oxygen species (ROS) [12]. Melanocytes and LESCs directly contact each other, which may suggest a supporting role for melanocytes in maintaining the function of the LN and LESCs [13]. Mesenchymal stem cells (MSCs), particularly CD90- and CD105-positive cells, seem to have close interactions with LESCs [14]. In confocal microscopy, these cells were detected adjacent to LESCs, which can be interpreted as evidence for this claim [15]. Additionally, several molecular signaling pathways were identified in this regard, as well as paracrine secretions and intercellular contact [14]. Cells at the base of the corneal limbus are positive for p63, Integrin β1 (CD29), and p75^NTR^ (CD271) [16].

#### 2.2.1. ECM of LN

The limbal epithelium basement membrane is composed of type IV collagen, α2 and β2-laminin, vitronectin, fibronectin, Integrin β1 (CD29), and tenascin C, which makes the structure of the limbal ECM completely distinct from that of the corneal stroma [17,18,19]. Overall, ECM components have various important interactions with niche cells. Hyaluronan (HA) is glycosaminoglycan, which makes up another component of the ECM and is produced by hyaluronan synthases (HASs), which have three types: HAS1, HAS2, and HAS3. Notably, all three types of HAS are expressed in the limbal area, and any defects in the expression of each enzyme can decrease the number of epithelial layers and speed of wound repair, as well as changes in the morphology of basal cells [20]. HA may have some role in the maintenance of the stem cell population, as one study showed that defective HAS2 leads to abnormalities in the compartment of LESCs [21]. Hence, HA not only acts as a bed to secure cells but also influences cellular behavior, making it an appropriate scaffold for use in cell or tissue transplantation [22].

#### 2.2.2. Genes and Proteins Implicated in LN Regulation

Several types of interactions have been described to regulate the activity and phenotype of LESCs, including direct cell–cell contact, paracrine signaling, autocrine signaling, and soluble factors [23]. Among these soluble factors, the Wnt signaling pathway is one of the key drivers of differentiation, proliferation, and quiescence of LESCs [24]. It has been shown that exposure of LESCs to high amounts of the Wnt6 ligand can lead to increased proliferation and lower expression of terminal differentiation markers of mature corneal epithelial cells [25]. Aside from the role of Wnt6 expression in the promotion of LESC self-renewal, it seems that the phenotype of LESCs is dependent on the Wnt7a–PAX6 axis [25,26]. Frizzled receptors are key components of Wnt signaling, and the Frizzled 7 (Fz7) receptor is the dominant type in the limbal area [27]. It has been reported that Fz7 receptor knockdown can lead to decreased marker expression and stemness of LESCs [27]. Therefore, manipulation of these signaling pathways could be of interest for clinical applications.

One of the other signaling pathways involved in LESC stemness is Jagged 1 (Jag1)-Notch signaling [28]. It has been reported that activation of this pathway can result in differentiation towards maturity of corneal epithelial cells and decreased LESC stemness. Therefore, therapies that inhibit Jag1-Notch signaling to enhance LESC stemness can be investigated in future studies.

The gene expression profile of inactive LESCs is completely different from that of mature corneal epithelial cells. Single-cell RNA sequencing (scRNA-seq) can help researchers to identify different genes involved in the differentiation and function of LESCs [29,30]. For example, Li and colleagues introduced TSPAN7 and SOX17 as critical factors in maintaining corneal epithelium homeostasis [31]. Additionally, SOX9 expression seems to have some role in the regulation of LESC activation or quiescence [32]. Furthermore, RUNX1, SMAD3, ATF3, ABCB5, H2AX, PBK, and Plk3 are among the other proteins and signaling pathways implicated in the modulation of the function and proliferation of LESCs [26,30,33]. These findings may justify future application of these proteins as potential markers to screen the success rate and outcomes of cultivated LESC transplantation [29]. Overall, these molecules show promising therapeutic applications for the near future, including increasing the transplantation success rate through effects on the self-renewal capacity and stemness of LESCs, introducing new drugs modulating the aforementioned pathways to medically manage partial-LSCD cases, and reprogramming corneal epithelial cells to transdifferentiate into an LESC-like phenotype, a dramatic shortcut to curing bilateral cases of LSCD [29].

## 3. LESCs’ Functions

### 3.1. Epithelial Maintenance

The turnover rate of the corneal epithelium is significantly high. Regeneration of the corneal epithelium occurs approximately every 2 weeks based on the XYZ hypothesis. In this theory, X stands for superficial movement of cells from the basal epithelium, Y is representative of centripetal migration of basal cells from the limbus, and Z represents damaged or desquamated lost cells [34]. The hypothesis claims that X + Y = Z, or, in other words, the loss of corneal cells is replenished by basal epithelial and limbal cells. Progenitor cells required for repopulation of the corneal epithelium are produced through division of LESCs located in the limbal basal layer. These progenitor cells, also known as transient amplifying cells (TACs), move centripetally and then superficially for terminal differentiation. In general, LESCs have a highly controlled division pattern: one daughter cell remains in the niche to maintain the LESC population while the other one differentiates into a TAC [35].

### 3.2. Epithelial Wound Healing

Several studies have reported the response and proliferation of limbal basal epithelial cells following large wounds [36,37]. However, small wounds can be resolved through enlargement of cell clusters of the central cornea [38]. It seems that limbal response starts with a latency period since movement and repopulation of the basal epithelium occurs about 8 h after wounding [23]. In addition to this key role of the limbus (e.g., proliferation of progenitor cells), it may induct a population pressure gradient to lead the migration of wound-edge basal epithelial cells into the wound bed [39].

## 4. LSCD

Various conditions have been implicated in causing LSCD due to severe damage to the LESCs or LN, among them ocular cicatricial pemphigoid (OCP), Stevens-Johnson syndrome (SJS), thermal or chemical burns, contact lenses, numerous ocular surgeries, local or systemic usage of 5-FU and MMC, and congenital aniridia [29,40]. LSCD is characterized by corneal opacity, neovascularization, and invasion of adjacent conjunctiva. LSCD interferes with corneal wound healing, resulting in subsequent complications such as persistent epithelial defect (PED), corneal ulcers, and even perforation [41]. Diagnosis of this entity is mainly clinical and based on slit-lamp examination findings. However, the gold-standard diagnostic method is impression cytology, which shows goblet or conjunctival cell markers in the corneal area. MUC5AC is used as a marker for goblet cells, and cytokeratin 7 and 13 identify conjunctival cells. Confocal microscopy and optical coherence tomography modalities are other useful diagnostic tools [42,43,44]. Details on this subject are outside of the scope of this review but are discussed in our previous review article [40].

From a microscopic point of view, inflammation is an inseparable part of LSCD, with alteration of several signaling cascades in both the cornea and limbus [45]. It has been reported that levels of pro-inflammatory cytokines (e.g., IL-1 and IL-6) and angiogenic molecules (e.g., vascular endothelial growth factor (VEGF)) are increased in the ocular surface of eyes with conjunctivalization [45]. Prolonged inflammatory conditions can result in unfavorable consequences, including angiogenesis, decreased expression of LESC markers, reduced colony-forming efficiency, and an altered ECM [14].

## 5. Limbal Stem Cell Transplantation

Several types of limbal transplantation are available based on the source (autologous or allogeneic) and preparation of the harvested tissue (direct or cultivated), including direct autologous transplantation, direct allogeneic transplantation, cultivated autologous transplantation, and cultivated allogeneic transplantation (Figure 2). A meta-analysis on the results of 40 studies was performed in 2020 to assess the outcomes of these 4 methods [46]. The results of this study agree with the superiority of autologous approaches in stabilizing the ocular surface; direct autologous transplantation and cultivated autologous transplantation had the highest success rates at 85.7% and 84.7%, respectively. The success rate of allogeneic methods was considerably lower: 57.8% for direct allogeneic transplantation and 63.2% for cultivated allogeneic methods. Direct autologous limbal transplantation was superior with regard to visual improvement [46]. Although allogeneic transplantation is one of the most successful approaches in the treatment of LSCD, one of the disadvantages of this method is the requirement of a long-term immunosuppressive regimen [47]. Hence, autologous methods are preferable. However, autologous approaches are not applicable in cases with bilateral LSCD [48]. To overcome this limitation, autologous cultivation methods were introduced.

### 5.1. Tissue Transplantation

LESC transplantation is required in severe cases of LSCD to replace the lost population of stem cells. The severity and extent of involvement are critical factors in choosing the appropriate approach and strategy. In unilateral cases with total involvement, the available options are conjunctival limbal autograft (CLAU) from the fellow eye and simple limbal epithelial transplantation (SLET) [14]. CLAU was introduced in the 1980s [49]. In this technique, two grafts of two clock hours each from the limbus and the adjacent rim of conjunctiva of the patient’s healthy fellow eye, are harvested and transplanted to the diseased eye. A success rate of 75% has been reported for CLAU [50]. SLET is a newer approach that was developed to minimize the risk of iatrogenic LSCD in the fellow healthy eye. In this method, only 1 small 2 × 2 mm (1-clock-hour) specimen from the patient’s normal eye is harvested and divided into smaller segments followed by transplantation to the diseased eye using an amniotic membrane and fibrin glue [51]. A success rate of 76% has been reported for autologous SLET in chemical injuries by Basu and colleagues [52]. In bilateral total LSCD, kerato-limbal allograft (KLAL) and living-related conjunctival limbal allograft (lr-CLAL) are available [53,54,55,56,57,58]. Overall, traditional approaches are based on harvesting a sample of functioning limbal tissue from a healthy eye [49]. More recently, approaches have utilized transplantation of cultivated and expanded LESCs (Table 1) [59].

### 5.2. LESC Culture and Expansion

In cases with unilateral involvement, autologous transplantation possesses the highest rate of success with a low risk of complications. However, the chance of developing iatrogenic LSCD in the healthy fellow eye is a concern [71]. This complication was frequently detected in rabbit models of autologous transplantation in which a 240° arc of limbal tissue was harvested [72,73]. On the other hand, harvesting tissues at a less than 90° arc was associated with transplantation failure [74,75]. So, it seems an intermediate size of tissue should be harvested to balance the risk of these two unfavorable outcomes.

To decrease the mentioned risk of iatrogenic damage, tissue-sparing methods were introduced. Over two decades ago, Pellegrini et al. [59] reported the first application of cultivated autologous transplantation, called cultivated limbal epithelial transplantation (CLET). In this technique, a tiny 2 × 2 mm section of limbal tissue is taken from the healthy eye, followed by ex vivo expansion of LESCs [76]. An amniotic membrane or a suspension is used as a scaffold to expand the harvested stem cells, which lasts 14–21 days [77,78]. A success rate of about 76% has been reported for CLET in chemical-burn-induced LSCD by Rama and colleagues [65]. Notably, it has been shown that the success rate of methods using cultivated stem cells is associated with the percentage of p63^+^ cells in cultures; Rama et al. reported a success rate of 78% for transplantations containing >3% p63^+^ cells. Meanwhile, this rate significantly decreases to 11% for transplantation of cultures with <3% p63^+^ cells [65]. Some studies reported graft survival might decrease over time, which could be related to the absence of a healthy niche [79,80]. In this regard, confocal microscopy has revealed that CLET is not capable of restoring the limbal niche [81]. It should be mentioned that CLET can be performed with autologous or allogeneic grafts. Allogeneic grafts are especially useful for bilateral cases of LSCD. A meta-analysis showed that the graft survival rate and visual improvement were equal for both autologous and allogeneic sources. However, autologous grafts are preferred as they do not require immunosuppression after surgery [82].

A technique offering the benefits of autologous transplantation (e.g., lack of immunosuppression and risk of disease transmission) in bilateral cases of LSCD without a suitable source of LESCs would be a valuable therapeutic tool. Hence, researchers began to use other stem cell lines to transdifferentiate into limbal stem cells, fulfilling this goal and need [29]. Historically, the first attempt in this line, in which the oral mucosa epithelium was cultivated and transplanted, was about two decades ago [48]. A brief review of the available non-limbal sources and relevant studies are provided below.

Oral mucosa epithelium: In 2004, the first usage of the oral mucosa epithelium in LSCD was reported [48]. In this study, six patients were enrolled, three of which were suffering from SJS and three of which had eyes with chemical burn. After 2–3 weeks of culture time, the prepared oral mucosa epithelium was implanted on an amniotic membrane scaffold with a supportive layer of fibroblasts and transplanted onto the diseased eyes. A success rate of about 70% has been reported for cultivated oral mucosal epithelial transplantation (COMET). Mild peripheral corneal neovascularization is the disadvantage of this technique. Moreover, the phenotype of oral epithelium remains unchanged after transplantation, leading to suboptimal visual outcomes due to this type being of a thicker and more opaque nature than the corneal epithelium [83,84].

Conjunctival epithelial cells: Similar to the previous method, conjunctival epithelial cells (CjECs) were used as another autologous source. After 18.5 months of follow-up, conjunctival epithelial transplantation (CjET) showed a 86% success rate in resolving conjunctivalization and corneal opacity [85]. Recovery of the corneal epithelium was approved using confocal microscopy, during which five to six layers of corneal epithelial cells with normal morphology were detected [85]. Overall, data on long-term survival with COMET and CjET grafts are limited.

Hair follicle epithelial stem cells: Follicular epithelial stem cells were reported to be positive for CD29 and CD271 [86]. Transdifferentiating of hair follicle epithelial stem cells to the corneal epithelium was studied in a murine model of LSCD [87]. After isolation and expansion, hair follicle epithelial stem cells were transferred to a medium similar to the limbal niche. Finally, these cells showed markers of corneal-epithelium-like cells, and an 80% success rate of transdifferentiation was observed. Further studies are required to generalize these results to human subjects.

Pluripotent stem cells: These cells are capable of forming a self-formed ectodermal autonomous multi-zone (SEAM), which contains cells of ectodermal lineage that mimic anterior and posterior eye development in vivo [88]. Hongisto et al. studied transdifferentiation of human pluripotent stem cells (PSCs) into human limbal stem cells and achieved over 65% LESCs in 24 days [89]. Additionally, they introduced a protocol to bank human-pluripotent-stem-cell-derived LESCs, which can facilitate further progress in these methods and similar research. Further research is required before implementation of this method in large-scale clinical trials. Recently, a team of scientists from Osaka University reported the results of the first ever trial on iPSC-based corneal transplantation [90]. They performed this trial successfully on four patients without any rejection or tumorigenicity.

Dental pulp: In a rabbit model of LSCD due to chemical injury, grafts containing human immature dental pulp stem cells (hIDPSCs) were transplanted into the limbal niche [91,92]. After 3 months, LESCs markers were detected on hIDPSCs, and the condition of the ocular surface was improved.

Umbilical cord stem cells: Human umbilical cord lining epithelial cells are another potential source for the management of LSCD. Animal models using this type of stem cell are available in the literature [93].

Embryonic stem cells: Human embryonic stem cells (hESCs) are pluripotent stem cells with the capability of differentiating into corneal and limbal epithelial cells [94]. Hence, application of these cells may be beneficial in LSCD. Although challenging, several in vitro models have been successfully used to differentiate hESCs into corneal-epithelial-like cells [95,96,97,98,99].

Amniotic membrane epithelial cells: It seems that expressed markers of amniotic membrane epithelial cells have a significant overlap with mesenchymal and embryonic stem cells. The other advantage of these cells is that they display immunomodulatory characteristics. In rabbit models, these cells have been successfully applied to treat LSCD [100,101].

Mesenchymal stem cells: this alternative source is separately discussed later.

Currently, most culture techniques are based on animal materials, which come with the risk of triggering the host immune system due to the transmission of non-human pathogens [102]. Nevertheless, studies using non-human reagents with acceptable outcomes are available [103,104,105]. Moreover, finding an optimum culture medium to simulate niche conditions in ex vivo is as important as using non-human reagents. In line with this concept, although the presence of supportive feeder cells is not necessary, they can significantly increase clonal efficiency through preserving cell–cell contact [106]. Monolayer irradiated or mitomycin-treated murine 3T3 fibroblasts (mitotically inactive) have been used previously as feeder cells to mimic a more suitable microenvironment. Meanwhile, currently, monolayer limbal mesenchymal cells and human-adipose-derived stem cells and bone marrow stromal cells are successfully applied in three-dimensional (3-D) culture systems [107,108,109].

## 6. LN Restoration

It seems that the pure transfer of LESCs without restoration of the LN does not lead to good long-term outcomes, especially in severe cases of LSCD [14,52,66,110]. Ongoing inflammation can act as a progressive destructive factor for remaining healthy stem cells. Hence, suppressing inflammation and recovery of LESCs and ECM function compose the foundations of niche restoration strategies (Figure 3) [14].

### 6.1. Bio-Scaffolds

After successful ex vivo expansion of LESCs, proper carriers should be used to transplant the grafts onto the targets.

#### 6.1.1. Amniotic Membrane

The most commonly used carrier in studies is the human amniotic membrane (HAM). The HAM, which has no vessels or nerves, contains various cytokines and growth factors, as well as collagen types I, III, IV, and V. So, this tissue has the potential to act as either a carrier for cell delivery or a scaffold for bioengineering [111,112]. Mimicking a niche-like environment for LESCs was previously proposed for the HAM [113]. Additionally, this matrix comprises anti-inflammatory, anti-fibrotic, and anti-angiogenic properties. The drawbacks of this agent are its low transparency and tensile strength and the risk of disease transmission [14,114]. Moreover, rapid digestion of the HAM after transplantation may eclipse its long-term outcomes [115].

#### 6.1.2. Fabrication of Bio-Active ECMs

Currently, several materials are used in the fabrication of bio-active ECMs, including decellularized corneas (human or animal) and purified/recombinant structural proteins such as collagen [116]. The process of corneal decellularization is performed via usage of ribonucleases, osmotic solutions, freeze thawing, and detergents to diminish the risk of antigenicity [117]. It should be mentioned that after this process, the ECM remains functional and structured with preservation of healing factors [118]. Decellularized porcine corneas were also transplanted to patients with corneal ulcers [119,120]. In these studies, the most suitable candidates were patients with stromal involvement but an intact epithelium. Hence, application of this method in cases of LSCD can be limited and lead to the development of an alternative option: hydrogel production through digestion of decellularized corneas [14,121,122]. In one study, a thermoresponsive hydrogel was fabricated from a decellularized porcine cornea after digestion using pepsin/HCl [123]. Numerous wound-healing factors were found in this hydrogel. Compatibility of this fabricated hydrogel with corneal cells makes it a proper cell delivery method for 3-D structures [124]. Moreover, further approaches are available to fabricate a bio-active hydrogel, including a silk-film-derived hydrogel with the ability to affect gene expression of the corneal epithelium, a cross-linked collagen hydrogel to substitute the corneal stroma, and a collagen-coupled polymer hydrogel that supports epithelial wound closure [125,126,127]. Regarding the purified/recombinant structural proteins, fabrication of bioengineered limbal crypts is achieved using collagen type I and cast molding [116]. The other approach is using 3-D printing via various bio-inks [128,129]. Collagen type I is the most common type in corneal structures. The biomechanics of collagen can be improved through several methods, such as cross-linking and plastic compression. Its suitable biomechanics, availability, and biocompatibility make collagen a suitable bio-scaffold [130,131].

#### 6.1.3. Others

Synthetic polymers and fibrin are among the other available options. Polyethylene glycol and polymethacrylate are constructed polymers with supportive roles in the cultivation of LESCs. However, they have not been studied in human trials yet [132,133]. Synthetic polymers offer several strengths, such as chemical stability, manipulability, and easy mass production [130]. Fibrin membrane, which is mostly composed of fibrinogen and thrombin, has a long history of safe application as a sealant in ophthalmology [134,135]. Fibrin can be prepared easily and showed an acceptable success rate in trials for LSCD.

### 6.2. Revitalization of Limbal Niche via Biological Factors

As mentioned before, signaling and cellular contacts are required for proper functioning of the limbal niche. Administration of exogenous factors can be used as an alternative to these signaling pathways [14].

#### 6.2.1. Blood-Derived Factors

Currently, ophthalmologists use autologous/allogeneic serum eye drops (ASEs) in routine practice for various ocular surface disorders, including dry eye disease (DED), PED, and corneal involvement following graft-versus-host disease (GVHD), and Sjögren disease [14]. ASEs are enriched with numerous cytokines and factors, such as TGF-β and EGF, as well as minerals and vitamins helpful in corneal epithelium maintenance and regeneration [136]. These properties justify the usage of ASEs in the management of ocular surface disturbances. Similarly, platelet-derived preparations, including platelet-rich plasma (PRP), platelet releasate (PR), and plasma rich in growth factors (PRGF), contain various growth factors, such as TGF, EGF, IGF-1, and pigment epithelium-derived factor (PEDF), highlighting the potential usefulness of platelet-derived products in limbal niche restoration [137].

#### 6.2.2. Bio-Active Soluble Factors/Cocktails

Different sources can be used to produce bio-active soluble factors/cocktails. One of these sources is amniotic membrane extract eye drops (AMEEDs). One study showed the enhancement of LESC functioning using in vivo cultivation with AMEEDs [138]. The other product extracted from the HAM is HC-HA/PTX3, which has shown to be effective in enhancement of self-renewal capacity of LESCs in 3-D culture systems through influencing the Wnt/BMP signaling pathway [139]. A similar function has been reported for PEDF, a soluble growth factor derived from human plasma, which activates the p38 MAPK and STAT3 signaling pathways [140].

The supernatant layer of in vitro cell cultivation is called secretome since it has all the secreted factors of those cells. Some studies have reported the mesenchymal stem cell (MSC) secretome can also promote LN and ocular surface regeneration. Additionally, MSC secretomes can lead to increased expression of the CD44 receptor and subsequent improvement in hyaluronic acid binding, which can decrease scar formation [141,142]. Other useful factors derived from MSCs include exosomes, which act in cell–cell contact. Corneal-MSC-derived exosomes can enhance wound repair capacity in animal corneas [143]. Additionally, corneal exosomes exhibit anti-inflammatory and immunomodulatory properties, which can address the pathophysiology of LSCD [144]. Furthermore, exosomes can act as a delivery vehicle [145].

Finally, conditioned media from limbal fibroblasts have shown promising results [146]. In an LSCD murine model, using limbal-fibroblast-conditioned media resulted in an increase in corneal-epithelial-like cells as well as lower density of conjunctival goblet cells [146].

### 6.3. Cell-Based Strategies

Currently, MSCs are the subject of many studies on LN and ocular surface reconstruction due to their formidable properties. Over half of a century has been passed since the initial isolation of these cells from bone marrow specimens [147]. The authors first noted the capability of MSCs in repairing bone defects [147]. The beneficial roles of MSCs are not limited to this finding, as their immunomodulatory functions have made them applicable in the treatment of autoimmune diseases and also organ transplantation [148]. Furthermore, they are also capable of producing ECMs in 3-D culture systems [149]. Application of MSCs in the management of chemical injuries, DED, and LSCD has been studied [150,151,152,153]. MSCs can be obtained from various sources, including bone marrow, adipose tissue and the HAM, limbus, and omentum [154,155,156,157,158]. It has been reported that bone-marrow-derived MSCs can decrease the level of inflammatory cytokines, oxidative stress species, and lipid peroxidation while increasing factors helpful for limbal niche restoration [159,160,161,162,163]. As discussed before, MSCs are one of the most important components found in a normal living LN. The properties that have been reported for limbal-derived MSCs are similar to those found for bone-marrow-derived ones [164,165]. MSCs also offer multiple advantages compared to limbal epithelial cells, including the ability to harvest from multiple tissues through a faster and cheaper process. Moreover, 100% of the MSCs in a transplant are stem cells [37]. In an animal model of chemical burn, local application of limbal-derived MSCs resulted in an increase in corneal transparency, a decreased epithelial defect, and attenuated corneal neovascularization [158]. Similarly, corneal MSCs secrete high levels of antiangiogenic factors [146]. Although data on the clinical application of MSCs are limited, the first clinical trial using allogeneic human-bone-marrow-derived MSCs reported a success rate of 76.5–85.7%, an efficacy similar to that of CLET [166]. Several routes are available to deliver the MSCs, including systemic topical, subconjunctival, sub-tenon, and intrastromal injection [167]. However, there is no general consensus on the optimal route for MSC delivery. Different routes of administration have specific drawbacks. The systemic route of administration may lead to a considerable rate of side effects, while a low number of cells may be delivered to the target site. On the topical route, the cells can be washed out, leading to a short period of cell retainment. In using a scaffold to transplant cells, the number of transferred cells is low, and the cost and risk of surgery should also be considered. Regarding the subconjunctival route, the best cell vehicle solution and cell concentration and also the number and location of injection are still unknown. Moreover, the volume of injection is limited. The intrastromal technique has more technical difficulties [152,159,168]. We conducted a clinical trial to evaluate the safety and maximally tolerated dose of locally delivered allogeneic MSCs. In this study, different doses of bone-marrow-derived MSCs were given using subconjunctival injections to evaluate safety as well as anatomical and functional results in adult cases of neurotrophic keratitis [169]. The results of the first three patients were reported in the annual ARVO 2022 meeting [170]. Overall, MSCs usage can be considered an emerging approach in the management of severe ocular surface disorders with promising results.

## 7. Conclusions

The presence of a competent limbal niche is completely necessary for proper functioning and homeostasis of LESCs. The limbal niche contains several components, including supportive cells, several signaling factors, neurovascular inputs, and a specialized ECM. Following severe acquired or hereditary injuries to the limbal niche resulting in LSCD, taking action to restore the niche is essential for therapeutic interventions to be successful. In addition to traditional LESC transplantation methods, regenerative approaches such as bio-scaffolds and cell-based therapies have attracted increasing attention. However, further clinical trials and human studies are required to incorporate these novel strategies into clinical practice.

## Figures and Tables

**Figure 1 bioengineering-10-00111-f001:**
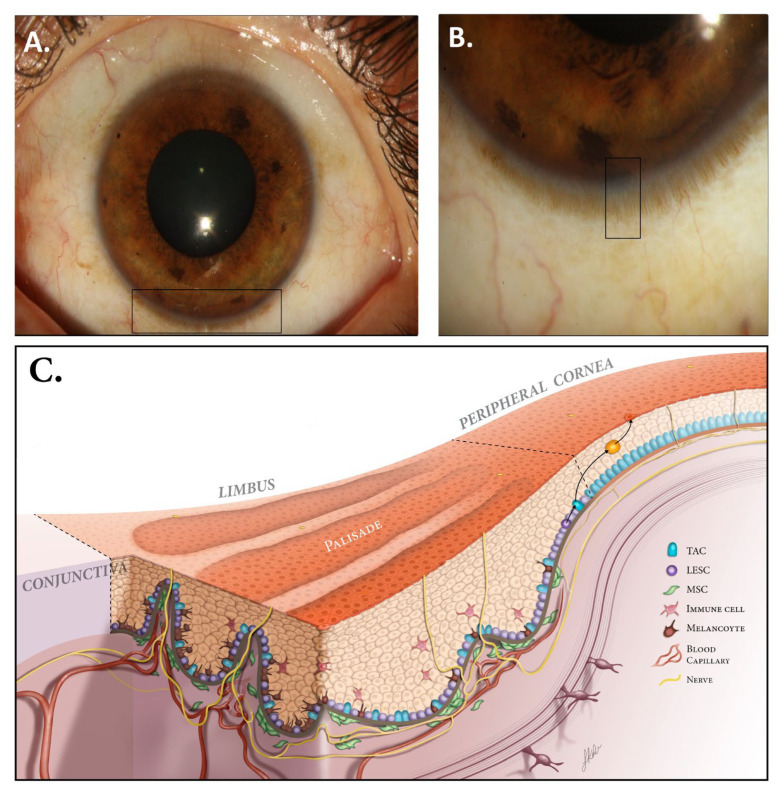
Normal ocular surface and limbus. (**A**) The corneoscleral limbus contains the palisades of Vogt (PVs), which have a length of 0.31 mm and a width of 0.04 mm and are typically more detectable on the superior and inferior sections of cornea. (**B**) Corneoscleral junction with magnification showing PVs. (**C**) The PVs contain different cells, such as melanocytes, mesenchymal stem cells, and immune cells. These cells, along with neurovasculature, provide growth factors, nutrients, and structural support to promote proper LESC proliferation and differentiation (LESC: limbal epithelial stem cell, TAC: transient amplifying cell, MSC: mesenchymal stem cell). Modified with permission from [14].

**Figure 2 bioengineering-10-00111-f002:**
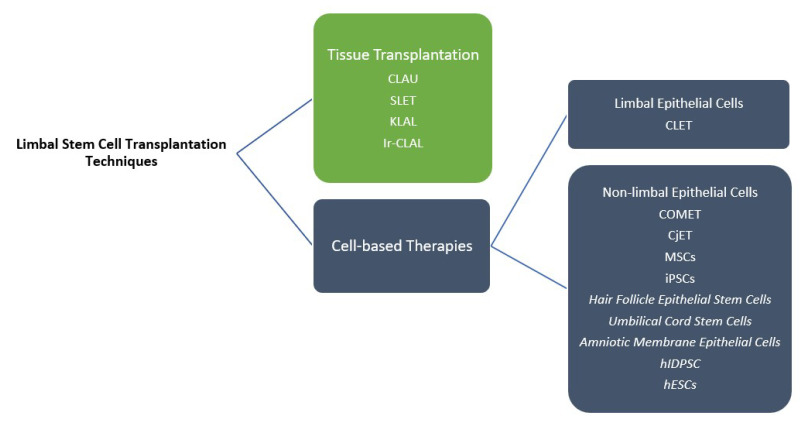
Different methods of limbal stem cell transplantation (CLAU: conjunctival limbal autograft, CLET: cultivated limbal epithelial transplantation, SLET: simple limbal epithelial transplantation, KLAL: kerato-limbal allograft, Ir-CLAL: living-related conjunctival limbal allograft, COMET: cultivated oral mucosal epithelial transplantation, CjET: conjunctival epithelial transplantation, MSCs: mesenchymal stem cells, iPSCs: induced pluripotent stem cells, hIDPSC: human immature dental pulp stem cells, hESCs: human embryonic stem cells). *Currently, only animal studies are available for methods written in italic format in the non-limbal epithelial cells box*.

**Figure 3 bioengineering-10-00111-f003:**
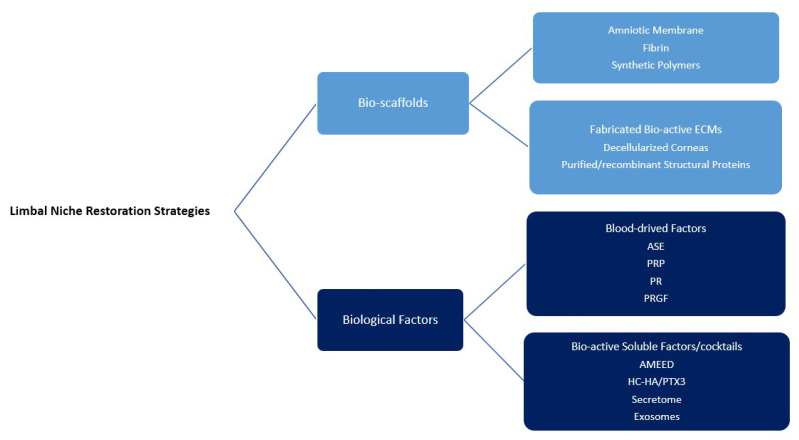
Different strategies and materials available for limbal niche restoration (ECM: extracellular matrix, ASE: autologous/allogeneic serum eye drops, PRP: platelet-rich plasma, PR: platelet releasate, PRGF: plasma rich in growth factors, AMEED: amniotic membrane extract eye drop).

**Table 1 bioengineering-10-00111-t001:** Advantages, disadvantages, and complications of limbal stem cell transplantation techniques (CLAU: conjunctival limbal autograft, CLET: cultivated limbal epithelial transplantation, SLET: simple limbal epithelial transplantation, KLAL: kerato-limbal allograft, Ir-CLAL: living-related conjunctival limbal allograft, COMET: cultivated oral mucosal epithelial transplantation, PED: persistent epithelial defect, LSCD: limbal stem cell deficiency).

Technique		Reference	Advantages	Disadvantages	Complications
CLAU		[60,61,62]	-Acceptable outcomes -Application of conjunctival patch in ocular surface reconstruction	Risk of iatrogenic LSCD	-Delayed epithelial healing -PED -Corneal perforation -Progressive conjunctival ingrowth
CLET		[63,64,65]	-Acceptable outcomes -Requirement of small donor tissue	-Expense -Technical difficulties -Risk of prion disease transmission via animal product usage during culture	-Postoperative hemorrhage under the graft -Infection -PED -Corneal perforation
SLET		[51,66]	-Acceptable outcomes -Requirement of small donor tissue	-Risk of donor tissue loss	-Focal recurrence of LSCD -Progressive conjunctivalization and symblepharon -Keratitis -PED
COMET		[67,68]	Applicable in bilateral cases	-Peripheral corneal neovascularization -Suboptimal visual outcomes	-PED -Corneal perforation -Glaucoma -Infection
Limbal allografts	lr-CLAL	[54,63]	-Applicable in bilateral cases -Utilizes a large conjunctival patch, which can be used in ocular surface reconstruction	-Requirement of immunosuppression regimen -Delayed epithelialization -Limited long-term success	-Rejection -Glaucoma -PED -Corneal melting and perforation -Graft-related issues -Infection -Posterior segment complications such as retinal detachment, vitreous hemorrhage, and cystoid macular edema
	KLAL	[63,69,70]	-Applicable in bilateral cases -Providing a larger number of LESCs compared to lr-CLAL		

## Data Availability

Not applicable.

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
