# Peer review of "Concise Review: Bioengineering of Limbal Stem Cell Niche"

_bioengineering, 2023, doi:10.3390/bioengineering10010111_

Round 1
Reviewer 1 Report
The paper, Limbal Stem Cell Niche Reconstruction, a Concise Review, is an excellent review on the limbal stem cell niche biology, function, disease, and therapy. This paper is well written and clearly describes the limbal stem cell niche and the critical importance of it in the biology of the healthy and diseased cornea. Furthermore, it describes current therapeutic options as well as therapies that are under investigation both in animals models and humans. This paper reviews the literature clearly and concisely in a subject matter that is clinically and biologically important.
Minor suggestions:
1. Figure 2 - the words in the yellow boxes are difficult to visualize, please consider changing the background color.
2. Line 433, page 10. Please cite the abstract that was presented at ARVO 2022 Investigative Ophthalmology & Visual Science June 2022, Vol.63, 91 – A0189.
This is an excellent review and would be important for both scientists and clinicians and therefore should be published.
Author Response
Dear Editor and Reviewers,
We are very grateful for thoughtful comments and recommendations. We wish to express our appreciation for the insightful comments. We have carefully addressed the editor and reviewer’s suggestions and in doing so feel the manuscript is substantially strengthened.
Reviewer #1:
- The paper, Limbal Stem Cell Niche Reconstruction, a Concise Review, is an excellent review on the limbal stem cell niche biology, function, disease, and therapy. This paper is well written and clearly describes the limbal stem cell niche and the critical importance of it in the biology of the healthy and diseased cornea. Furthermore, it describes current therapeutic options as well as therapies that are under investigation both in animals’ models and humans. This paper reviews the literature clearly and concisely in a subject matter that is clinically and biologically important.
Figure 2 - the words in the yellow boxes are difficult to visualize, please consider changing the background color.
Dear reviewer, thank you so much for your valuable comments; the mentioned figure has been updated.
- 2. Line 433, page 10. Please cite the abstract that was presented at ARVO 2022 Investigative Ophthalmology & Visual Science June 2022, Vol.63, 91 – A0189.
This is an excellent review and would be important for both scientists and clinicians and therefore should be published.
Thanks for your comment; the mentioned reference has been added to the manuscript.

Reviewer 2 Report
Main Comment
The review is both complete and concise. It is a good synthesis of the different therapeutic strategies for LSCD. But the same team has already published 3 similar reviews in 2019, 2021 and 2022 (Yazdanpanah et al., Ocul Surf, 2019 (PMID: 30633966), Elhusseiny et al., Stem Cells Transl Med (PMID: 35303110 ), Amin et al., Vision, 2021 (PMID: 34698278). Similar reviews had been also published by other teams ( Michel Haagdorens et al., Stem cells int, 2016 (PMID 26788074), Ramachandran et al., Stem cell Transl Med, 2014, (PMID: 25205842), O'Callaghan et al., Stem cells, 2011, (PMID: 21997829)). It is necessary to differentiate from all previous similar reviews. For example, you can simplify the pathophysiology part of LSCD, and focus on the different techniques to treat LSCD by giving tables, explanatory figures in order to highlight the difference (strategic and/or technical), advantages and disadvantages of each technique…
Others comments
1-2: Title/ Limbal stem cell niche reconstruction; concise review
1- It would be better: Concise Review: Limbal stem cell niche reconstruction
2- I’m not sure that’s an appropriate term: Limbal stem cell niche reconstruction. Do we have any (anatomical/histological) evidence that the limbal stem cell niche has been rebuilt after these therapies?
58: Please be sure about the size of palisades of Vogt: 0.31mm length and 0.04 mm width by citing the references.
Fig 1:
1- It's better to use the real photos instead of Fig 1A and B to indicate the PV, as it is clearer for new students/researchers in the field
2- Please indicates the limbal epithelial crypts.
3- Figure 1C gives us the impression that the PV represent areas with more cell layers? Is this true? Which cell/tissue structures correspond to the PV? Is there a relationship between PV and limbal crypts?
114-128 : This paragraph does not seem to fit the “2.2.2 Solube factors of LN” section.
189-190: please give several references for KLAL and CLAL
Fig 2: Are there therapies based on bio-scaffolds without adding cells? If yes, please differentiate these 2 types of therapies: Bio-scaffolds with and without adding cells; if no, please put "bio-scaffolds" under "cell-based therapies".
405-436: It would be better to specify the 3 applications of MSC: Topical, Sub-tenon and intravenous. Does the Topical method need a Bio-scaffold? What are the advantages and disadvantages of each technique? Are they applied in combination with cell-based therapies/tissue transplantation or alone?...
Author Response
Dear Editor and Reviewers,
We are very grateful for thoughtful comments and recommendations. We wish to express our appreciation for the insightful comments. We have carefully addressed the editor and reviewer’s suggestions and in doing so feel the manuscript is substantially strengthened.
Reviewer #2:
- The review is both complete and concise. It is a good synthesis of the different therapeutic strategies for LSCD. But the same team has already published 3 similar reviews in 2019, 2021 and 2022 (Yazdanpanah et al., Ocul Surf, 2019 (PMID: 30633966), Elhusseiny et al., Stem Cells Transl Med (PMID: 35303110), Amin et al., Vision, 2021 (PMID: 34698278). Similar reviews had been also published by other teams (Michel Haagdorens et al., Stem cells int, 2016 (PMID 26788074), Ramachandran et al., Stem cell Transl Med, 2014, (PMID: 25205842), O'Callaghan et al., Stem cells, 2011, (PMID: 21997829)). It is necessary to differentiate from all previous similar reviews. For example, you can simplify the pathophysiology part of LSCD, and focus on the different techniques to treat LSCD by giving tables, explanatory figures in order to highlight the difference (strategic and/or technical), advantages and disadvantages of each technique…
Dear reviewer, thanks for your comments; Compared to the previous works, limbal niche is described in a more detailed and cutting-edge discussion. Based on your useful comment, Table 1 is drawn showing the advantages, disadvantages, and complications of different techniques.
- Title/ Limbal stem cell niche reconstruction; concise review
1- It would be better: Concise Review: Limbal stem cell niche reconstruction
2- I’m not sure that’s an appropriate term: Limbal stem cell niche reconstruction. Do we have any (anatomical/histological) evidence that the limbal stem cell niche has been rebuilt after these therapies?
We thank the reviewer for this comment; the title is changed to concise review: bioengineering of the limbal stem cell niche.
- Please be sure about the size of palisades of Vogt: 0.31mm length and 0.04 mm width by citing the references.
Thank you very much; the aforementioned phrase has been rechecked and confirmed. You may find more details at https://europepmc.org/backend/ptpmcrender.fcgi?accid=PMC1298638&blobtype=pdf.
- Fig 1: It's better to use the real photos instead of Fig 1A and B to indicate the PV, as it is clearer for new students/researchers in the field.
Please indicates the limbal epithelial crypts.
Figure 1C gives us the impression that the PV represent areas with more cell layers? Is this true? Which cell/tissue structures correspond to the PV? Is there a relationship between PV and limbal crypts?
Thanks for your comments; real images are substituted with schematic images. Limbal epithelial crypts are not visible in the image. For better understanding, some explanations and references are added to the text of the manuscript. Readers will be able to see the original papers for more details.
Limbal epithelial crypts are projections from the undersurface of the limbal epithelium into the stroma. These structures could be parallel or perpendicular to the Palisades of Vogt. Focal stromal projections are finger-shaped projections of stroma containing a central blood vessel, which extend upward into the limbal epithelium.
- 114-128: This paragraph does not seem to fit the “2.2.2 Solube factors of LN” section.
We thank the reviewer for this comment; the title of subheading has been changed to implicated genes and proteins in LN regulation to fit the materials.
- 189-190: please give several references for KLAL and CLAL.
Thank you very much; done!
- Fig 2: Are there therapies based on bio-scaffolds without adding cells? If yes, please differentiate these 2 types of therapies: Bio-scaffolds with and without adding cells; if no, please put "bio-scaffolds" under "cell-based therapies".
Thank you; the figure has been revised.
- 405-436: It would be better to specify the 3 applications of MSC: Topical, Sub-tenon and intravenous. Does the Topical method need a Bio-scaffold? What are the advantages and disadvantages of each technique? Are they applied in combination with cell-based therapies/tissue transplantation or alone?
Thank you very much; in fact, several routes are available to deliver the MSCs including, systemic topical, subconjunctival, sub-tenon, and intrastromal injection. However, there is no general consensus on the route of MSC delivery. Different routes of administration have their specific drawbacks; systemic route of administration may lead to a considerable rate of side effects, meanwhile a low number of cells may be delivered to the target site. On topical route the cells can be washed out, leading to a low time of cell retainment. Using a scaffold to transplant cells, the number of transferred cells is low and also the cost and risk of surgery should be considered. Regarding the subconjunctival route, the best cell vehicle solution and cell concentration and also the number and location of injection are still unknown. Moreover, the volume of injection is limited. Intrastromal technique has more technical difficulties.

Reviewer 3 Report
This review article is a well-reviewed and reader-friendly manuscript. A few additional corrections are requested.
Major points
1) Please add Integrin B1 (CD29), which is expressed in the basement membrane of the corneal limbus, to the "ECM of LN" section from Line 82.
2) CD90 and CD105 are mesenchymal stem cell markers and their association with LESCs was introduced (Line 68). p63 was introduced (Line 238-240). p63 was introduced (Line 238-240). The association between follicular epithelial stem cells and LESCs was described (Line 273). On the other hand, follicular epithelial stem cells were reported to be positive for CD29 and CD271 (Inoue K, Aoi N, Sato T, Yamauchi Y, Suga H, Eto H, Kato H, Araki J, Yoshimura K. Differential expression of Lab Invest. 2009 Aug;89(8):844-56. doi: 10.1038/labinvest.2009.48. Epub 2009 Jun 8. PMID: 19506554.)
Therefore, please add a reference to add that cells at the base of the corneal limbus are positive for p63, Integrin beta1 (CD29: epithelial/mesenchymal stem cell marker), and p75NTR (CD271: epithelial/mesenchymal stem cell marker) (Yamamoto N, Hirano K, Kojima H, Sumitomo M, Yamashita H, Ayaki M, Taniguchi K, Tanikawa A, Horiguchi M. Cultured human corneal epithelial stem/progenitor cells In Vitro Cell Dev Biol Anim. 2010 Oct;46(9):774-80. doi: 10.1007/s11626-010-9344-9. Epub 2010 Sep 16: 20844981.)
It would be easier for readers to understand the relevance of these markers if it were stated in the text that p63 is positive in basal cells of the corneal ring and that epithelial and mesenchymal stem cell markers such as Integrin beta1 (CD29) and p75NTR (CD271) are positive. As a result, reviewers will appreciate the review article more.
3) The words self-formed ectodermal autonomous multi-zone (SEAM) should be listed and introduced (Line 283).
The reference 73 is incorrect.: Hayashi R, Ishikawa Y, Sasamoto Y, Katori R, Nomura N, Ichikawa T, Araki S, Soma T, Kawasaki S, Sekiguchi K, Quantock AJ, Tsujikawa M, Nishida K. Co-ordinated ocular development from human iPS cells and recovery of corneal function. Nature. 2016 Mar 17;531(7594):376-80. doi 10.1038/nature17000. epub 2016 Mar 9. PMID: 26958835.
Minor point
1) Please increase the image quality so that the text in Figure 2 is easier to read (Line 193).
Author Response
Dear Editor and Reviewers,
We are very grateful for thoughtful comments and recommendations. We wish to express our appreciation for the insightful comments. We have carefully addressed the editor and reviewer’s suggestions and in doing so feel the manuscript is substantially strengthened.
Reviewer #3:
- This review article is a well-reviewed and reader-friendly manuscript. A few additional corrections are requested.
Please add Integrin B1 (CD29), which is expressed in the basement membrane of the corneal limbus, to the "ECM of LN" section from Line 82.
Thank you for your comment: done!
- CD90 and CD105 are mesenchymal stem cell markers and their association with LESCs was introduced (Line 68). p63 was introduced (Line 238-240). The association between follicular epithelial stem cells and LESCs was described (Line 273). On the other hand, follicular epithelial stem cells were reported to be positive for CD29 and CD271 (Inoue K, Aoi N, Sato T, Yamauchi Y, Suga H, Eto H, Kato H, Araki J, Yoshimura K. Differential expression of Lab Invest. 2009 Aug;89(8):844-56. doi: 10.1038/labinvest.2009.48. Epub 2009 Jun 8. PMID: 19506554.)
Therefore, please add a reference to add that cells at the base of the corneal limbus are positive for p63, Integrin beta1 (CD29: epithelial/mesenchymal stem cell marker), and p75NTR (CD271: epithelial/mesenchymal stem cell marker) (Yamamoto N, Hirano K, Kojima H, Sumitomo M, Yamashita H, Ayaki M, Taniguchi K, Tanikawa A, Horiguchi M. Cultured human corneal epithelial stem/progenitor cells In Vitro Cell Dev Biol Anim. 2010 Oct;46(9):774-80. doi: 10.1007/s11626-010-9344-9. Epub 2010 Sep 16: 20844981.)
It would be easier for readers to understand the relevance of these markers if it were stated in the text that p63 is positive in basal cells of the corneal ring and that epithelial and mesenchymal stem cell markers such as Integrin beta1 (CD29) and p75NTR (CD271) are positive. As a result, reviewers will appreciate the review article more.
Thank you very much; the aforementioned references have been added to the text.
- The words self-formed ectodermal autonomous multi-zone (SEAM) should be listed and introduced (Line 283).
The reference 73 is incorrect.: Hayashi R, Ishikawa Y, Sasamoto Y, Katori R, Nomura N, Ichikawa T, Araki S, Soma T, Kawasaki S, Sekiguchi K, Quantock AJ, Tsujikawa M, Nishida K. Co-ordinated ocular development from human iPS cells and recovery of corneal function. Nature. 2016 Mar 17;531(7594):376-80. doi 10.1038/nature17000. epub 2016 Mar 9. PMID: 26958835.
Thank you for your comment, the correction has been made. Reference 73 has been rechecked and confirmed.
- Please increase the image quality so that the text in Figure 2 is easier to read (Line 193).
Thanks for your comment; the figure has been updated.

Round 2
Reviewer 2 Report
Thank you for your speed and efforts. Your team is successful and well advanced in the field. I give you my approval for publication and wish you the best innovative research results in 2023.